# CCD-Rice: A long-term paddy rice distribution dataset in China at 30 m resolution

Ruoque Shen [1], Qiongyan Peng [1], Xiangqian Li [1], Xiuzhi Chen [1], and Wenping Yuan [2]

[1]School of Atmospheric Sciences, Sun Yat-sen University, Zhuhai 519082, Guangdong, China

[2]Institute of Carbon Neutrality, Sino-French Institute for Earth System Science. College of Urban and Environmental Sciences, Peking University, Beijing 100871.China

*Correspondence to*: Wenping Yuan (yuanwp@pku.edu.cn)

**Abstract**. As one of the most widely cultivated grain crops, paddy rice is a vital staple food in China and plays a crucial role in ensuring food security. Over the past decades, the planting area of paddy rice in China has shown substantial variability. Yet, there are no long-term high-resolution rice distribution maps in China, which hinders our ability to estimate greenhouse gas fluxes and crop production. This study developed a new optical satellite-based rice mapping method using a machine learning model and appropriate data preprocessing strategies to address the challenges of cloud contamination and missing data in optical remote sensing observations. This study produced CCD-Rice (China Crop Dataset-Rice), the first high-resolution rice distribution dataset in China from 1990 to 2016. Based on 397,414 validation samples, the overall accuracy of the distribution maps in each provincial administrative region averaged 89.89 %. Compared with 20,544 county-level statistical data, the coefficients of determination ($R^2$) of single- and double-season rice in each year averaged 0.85 and 0.78, respectively. The distribution maps can be obtained at https://doi.org/10.57760/sciencedb.15865 (Shen et al., 2024a).

**Keywords**: rice; single-season rice; double-season rice; Landsat; crop mapping

## 1. Introduction

Paddy rice (*Oryza sativa*) is one of the most critical crops in the world, accounting for 8 % of global food production in 2021, and is a staple food for more than 50 % of the world population (Elert, 2014; FAO, 2023). However, rice cultivation is a major consumer of freshwater resources and a significant source of methane emissions, a potent greenhouse gas (Bouman et al., 2007; Mohammadi et al., 2020; Zhang et al., 2020). In addition, the spatial distribution of rice cultivation has changed significantly over the past few decades (Jiang et al., 2019; Liu et al., 2013). Therefore, the long-term identification of paddy rice is very important for food security, water resource management, and climate change research.

The satellite data used for rice mapping can be categorized into two types: optical remote sensing data and synthetic aperture radar (SAR) data. While optical remote sensing data from satellites such as the Moderate-resolution Imaging Spectroradiometer (MODIS), Landsat, and Sentinel-2 have been widely used, they face significant challenges in data quality (Li and Chen, 2020). Cloud cover frequently obstructs the acquisition of ground surface reflectance, particularly in major rice-





growing regions (Jiang et al., 2021; Shen et al., 2023a). This issue is especially pronounced for high-resolution satellites like

the Landsat series and Sentinel-2, which have spatial resolutions of 10 to 30 meters but long revisit periods of 16 and 5 days,

respectively (Rahimi and Jung, 2024; Sudmanns et al., 2020). For instance, in southern China, where rice is extensively

cultivated, annual averages of cloud-free Landsat observations were fewer than eight between 1984 and 2017 (Zhou et al.,

2019). Such sparse observations pose challenges to rice mapping studies, as clouds can severely affect classification accuracy

(Dong et al., 2016; Shen et al., 2023a). On the other hand, MODIS has a relatively high revisit frequency, with two satellites,

Terra and Aqua, providing observations every one to two days, enabling relatively dense temporal coverage for rice mapping

(Clauss et al., 2016; Han et al., 2022; Xiao et al., 2005, 2006). However, its coarse spatial resolution of 250 to 1000 m leads

to significant confusion in regions dominated by smallholder fields due to the issue of mixed pixels (Fritz et al., 2015; Tan et

al., 2006; Yan et al., 2016). Although SAR data from Sentinel-1 can overcome cloud-related limitations and provide all-weather

images, it suffers from salt-and-pepper noise compared to optical images and was not widely available before Sentinel-1A's

launch in 2014 (Nguyen et al., 2016; Oguro et al., 2001; Oliver and Quegan, 2004; Sun et al., 2023; Veloso et al., 2017).

Consequently, achieving long-term, high-resolution rice mapping still needs to face the challenge posed by poor-quality optical

remote sensing data.

Existing rice mapping methods also face limitations when dealing with poor-quality optical data. There are two main

approaches for rice mapping: phenology-based and machine-learning methods. Phenology-based methods rely on the distinct

phenological characteristics of rice, particularly the flooding signal during transplantation (Han et al., 2021; Nguyen et al.,

2016; Pan et al., 2021b; Phan et al., 2018; Xiao et al., 2005, 2006). However, the short duration of flooding signals, typically

lasting only a few weeks, makes these methods particularly sensitive to data quality issues (Shen et al., 2023a). Missing values

during the crucial phenological periods are likely to result in incorrect identification, thereby reducing classification accuracy

(Dong et al., 2016). As a result, existing high-resolution rice mapping studies either rely on SAR data or only focus on regions

with fewer clouds, such as northeastern China (Dong et al., 2015, 2016; Hu et al., 2023; Xu et al., 2023; Zhang et al., 2023b).

While machine learning methods, including support vector machine (SVM), random forest (RF), and deep learning approaches,

can achieve high accuracy (Mansaray et al., 2020; Sun et al., 2023; Tian et al., 2023; Waleed et al., 2022; You et al., 2021).

However, they typically require large volumes of training samples and face challenges with model transferability across

different years (Valero et al., 2016). These limitations underscore the urgent need for novel methods that can effectively handle

poor-quality optical data while ensuring reliable rice mapping accuracy for long-term rice mapping.

China is the world's largest rice producer, and, until 2017, rice was the most widely cultivated grain crop in the country

(FAO, 2023; National Bureau of Statistics of China, 2023). Rice is also one of the most important staple foods in China,

consumed by more than two-thirds of the population, especially in southern China, where it can account for more than 80 %

of cereal intake (Zhao et al., 2023). Although there have been many previous studies on mapping rice in China, a nationwide, long-term, high-resolution rice map is still lacking. Some studies, such as those by Pan et al., (2021b) and Shen et al., (2023a), have produced nationwide distribution maps of double- and single-season rice in China, respectively. However, due to limitations in the quality of the remote sensing data, both studies covered only recent years (2016–2020 and 2017–2022, respectively). To address this gap, this study focuses on mapping rice distribution before 2017 and tackling the challenge of

poor-quality remote sensing data. Specifically, this study intends to (1) develop a new optical satellite-based rice mapping method; (2) produce high-resolution distribution maps of single- and double-season rice in China from 1990 to 2016; (3) evaluate the accuracy of the results and analyze changes in rice cultivation patterns.

## 2. Data and methods

### 2.1 Study area

Rice is cultivated in most of the provincial administrative regions of China. The study area for this research was selected to include 25 provincial administrative regions in the eastern monsoon region of mainland China. The proportion of rice planting area in these 25 provincial administrative regions ranged from 99.60 % to 99.74 % of the total rice planting area in mainland China from 1990 to 2016 (https://data.stats.gov.cn). Due to differences in cloud cover and rice calendar in each provincial administrative region, the study area was divided into four subregions (Fig. 1). Subregion I is located in northern

China and includes Heilongjiang, Jilin, Liaoning, Hebei, Inner Mongolia, Ningxia, and Tianjin. Here, only single-season rice is cultivated, and optical satellite images are less affected by cloud cover due to relatively low precipitation. Subregion II, includes Jiangsu, Sichuan, Yunnan, Chongqing, Guizhou, Henan, Shanghai, Shaanxi, and Shandong. Subregion II is also planted with only single-season rice, but experiences more precipitation, leading to poorer quality of optical remote sensing data than in subregion I. Subregion III, which includes Hunan, Jiangxi, Hubei, Anhui, and Zhejiang, cultivates both single-

and double-season rice. Subregion IV, which is warmer, allows for early rice cultivation than in subregion III and includes Guangxi, Fujian, Guangdong, and Hainan. Among these, Guangxi and Fujian cultivate both single- and double-season rice, while Guangdong and Hainan cultivate only double-season rice.

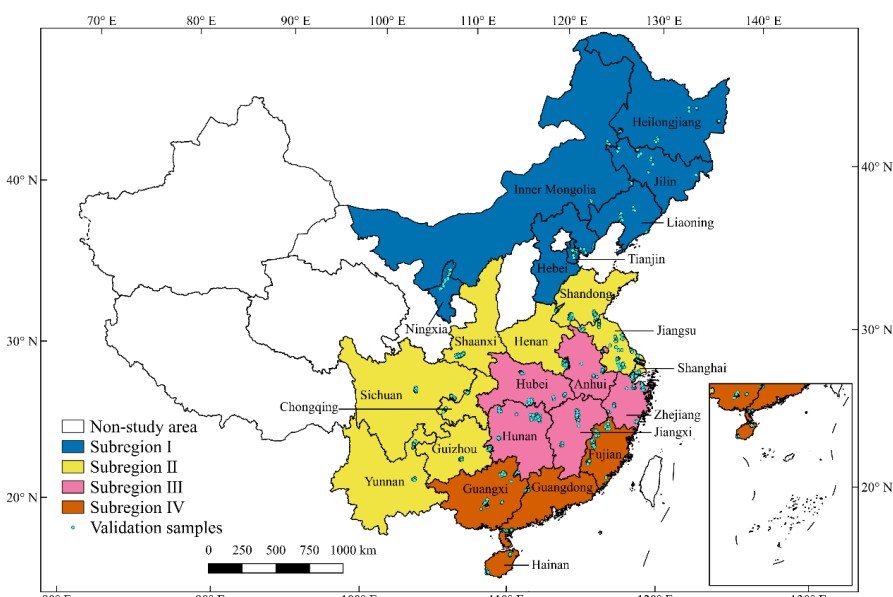

**Figure 1: Study area and validation samples. The study area is divided into four subregions (shaded areas). The green dots indicate the centers of validation sample polygons.**

## 2.2 Data

### 2.2.1 Satellite data and landcover data

The satellite data used for rice mapping in this study were sourced from the Landsat Collection 2 Level-2 Science Products, distributed by the United States Geological Survey (USGS). This product represents atmospherically corrected surface reflectance for the Landsat series. This study used band B5 of Landsat 5 and 7 as well as band B6 of Landsat 8, which both correspond to shortwave infrared 1 (SWIR1), with wavelength ranges of 1.55 to 1.75 μm and 1.566 to 1.651 μm, respectively. SWIR1 is sensitive to land surface water, and as such can capture the unique flooding signal during rice transplanting. Previous studies have demonstrated its effectiveness for rice mapping (Shen et al., 2023a). In addition, Landsat 7 data were not used after 2002 due to the failure of its Scan Line Corrector (SLC) on May 31, 2003. However, in 2012, Landsat 7 data had to be used despite the SLC malfunction, as Landsat 5 had been retired, and Landsat 8 had not yet been launched, leaving Landsat 7 as the only available Landsat satellite that year. The SWIR1 reflectance used for model training in this study was obtained from Landsat 8 and 9 from Landsat Collection 2 Level-2 Science Products as well as Sentinel-2 data provided by the European Space Agency (ESA).

The Quality Assessment (QA) band of Landsat data was used to eliminate the effect of clouds on Landsat images, while the Sentinel-2 Cloud Probability (S2C) product (https://developers.google.com/earth-engine/datasets/catalog/COPERNICUS_S2_CLOUD_PROBABILITY) was used to exclude cloud-covered pixels from



Sentinel-2 images. Pixels with a cloud probability greater than 50 % in the S2C product were considered to include cloud cover and were subsequently removed. The cloud-removed images were further composited to a median 8-day temporal resolution. For missing values in the time series due to cloud cover or observation frequency, this study did not use interpolation or other

approaches to fill them, but uniformly set them to zero. Cloud removal and compositing were performed on the Google Earth Engine (GEE) platform (Gorelick et al., 2017).

In this study, the China land cover dataset (CLCD) product produced by Yang and Huang (2021, 2023) was used to exclude non-cultivated pixels. This product maps the land cover of China from 1985 to 2022 at a spatial resolution of 30 m and was produced using random forest (RF). The user's accuracy of cultivated land is 78.43 %.

**2.2.2 Rice distribution maps of recent years**

The training samples used in this study were extracted from two rice distribution maps of recent years: the distribution map of single-season rice in China from 2017 to 2022 produced by Shen et al. (2023a, b) and the distribution map of double-season rice in China from 2016 to 2020 produced by Pan et al. (2021a, b). The average overall accuracies of these two products over their studied provincial administrative regions were 85.23 % and 91.17 %, respectively. For provinces where only single-

season rice or only double-season rice was cultivated, this study used the distribution maps for all years of two products, respectively. For provinces where both single- and double-season rice were cultivated, this study used only the common years of the two products, i.e., 2017 to 2020.

**2.2.3 Validation sample and agricultural statistical data**

The validation data used in this study consisted of validation samples and agricultural statistical data. The validation

samples were visually interpreted from the very high-resolution images of Google Earth. The availability of imagery suitable for visual interpretation is limited by the scarcity of historical images from earlier years on Google Earth in China and the fact that early images tend to be for urban areas rather than rural areas. Therefore, instead of collecting validation samples across all study years, this study selected data from only two to four years in each provincial administrative region. We collected a total of 3501 polygons including 1826 and 867 polygons of single- and double-season rice field, respectively, and 808 polygons

of other cover types (non-rice crops, natural vegetation, built-up areas, water bodies etc.) from 2002 to 2016, and further converted them into a total of 397,414 validation samples with a 30-m spatial resolution, including 191,434 single-season rice samples, 11,456 double-season rice samples, and 194,524 samples of other cover types (Fig. 1 and Table 1).

**Table 1: Years and number of validation samples in each provincial administrative region**

| Province | Years of samples | Number of samples | | |
|---|---|---|---|---|
| | | SR | DR | Other |



| | | SR | DR | |
|---|---|---|---|---|
| Heilongjiang | 2010, 2011, 2016 | 67329 | 0 | 41571 |
| Jilin | 2007, 2015 | 69472 | 0 | 40611 |
| Liaoning | 2011, 2015 | 2484 | 0 | 5621 |
| Hebei | 2008, 2015 | 1550 | 0 | 2492 |
| Inner Mongolia | 2013, 2015 | 2559 | 0 | 2543 |
| Ningxia | 2010, 2015 | 3082 | 0 | 3769 |
| Tianjin | 2002, 2006, 2011, 2014 | 4888 | 0 | 7304 |
| Jiangsu | 2004, 2011, 2013, 2014 | 6396 | 0 | 8656 |
| Sichuan | 2005, 2015 | 1896 | 0 | 1950 |
| Yunnan | 2005, 2013 | 2568 | 0 | 2190 |
| Chongqing | 2005, 2016 | 524 | 0 | 776 |
| Guizhou | 2012, 2015 | 899 | 0 | 1018 |
| Henan | 2010, 2016 | 1506 | 0 | 2937 |
| Shanghai | 2004, 2014 | 13380 | 0 | 26330 |
| Shaanxi | 2005, 2014, 2015 | 2853 | 0 | 2260 |
| Shandong | 2010, 2013 | 3409 | 0 | 6157 |
| Hunan | 2013, 2015 | 806 | 773 | 1846 |
| Jiangxi | 2006, 2012 | 561 | 1424 | 1976 |
| Hubei | 2010, 2015 | 1011 | 123 | 3036 |
| Anhui | 2003, 2013 | 897 | 593 | 1269 |
| Zhejiang | 2003, 2013 | 1992 | 977 | 3918 |
| Guangxi | 2011, 2013, 2016 | 582 | 1576 | 2535 |
| Fujian | 2013, 2015 | 790 | 209 | 1153 |
| Guangdong | 2010, 2012 | 0 | 5291 | 21922 |
| Hainan | 2009, 2014 | 0 | 490 | 684 |

SR and DR mean single- and double-season rice, respectively.


In this study, the planting area of single- and double-season rice in each provincial administrative region was collected from the following website: https://data.stats.gov.cn. This study also collected the rice planting area at the county level from the statistical yearbooks of provinces or cities. However, since it is difficult to trace statistical yearbooks back to 1990, and in some places the statistical yearbooks did not record the rice planting area, we were unable to collect complete statistics in all

the years for all the county-level administrative regions. Furthermore, due to discrepancies between administrative divisions and statistical reporting, as well as changes in administrative divisions, some statistical data recording planting areas do not align with current or actual jurisdiction, making them unusable. Ultimately, this study was able to collect a total of 20,544 records at the county level from 1991 to 2016 (Fig. 2).



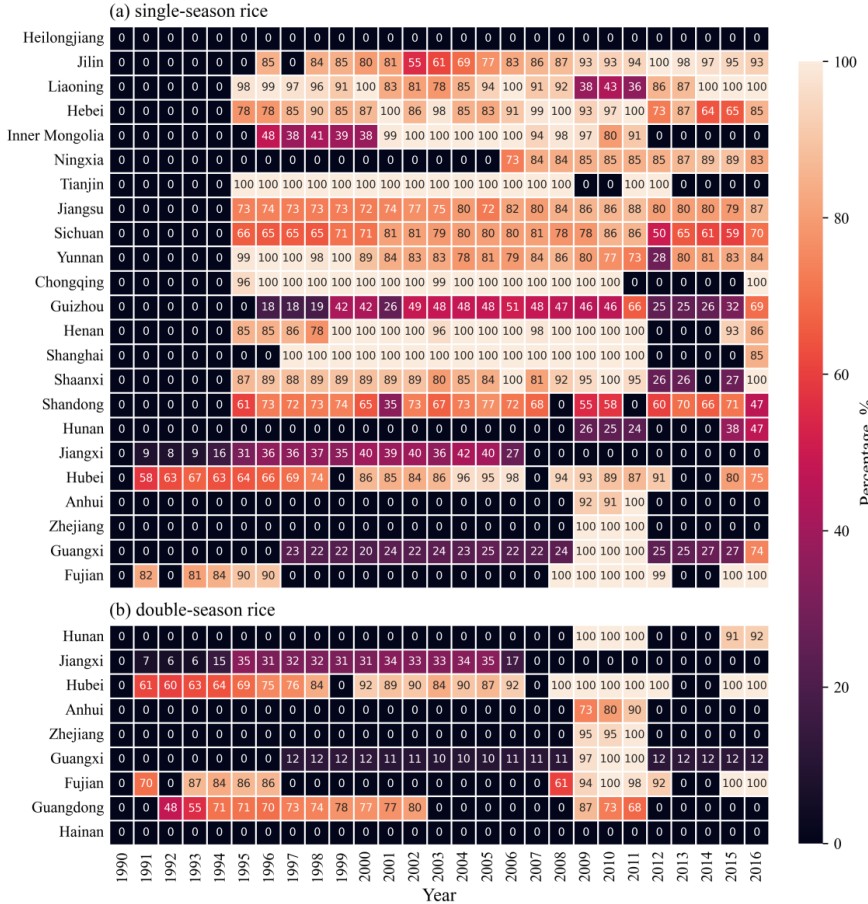

Figure 2: Percentage of the collected statistical planting areas relative to the province-level planting area in each provincial administrative region from 1990 to 2016.

### 2.2.4 Existing products

There are other studies that have produced long-term distribution maps of rice at regional, national, or continental scales. Our product will be compared with the following four open-access products: (1) *NEAsia_Rice* product, which produced the rice distribution in Northeast China from 2000 to 2017 at 500-m resolution using MODIS imagery and a phenology-based method (Xin et al., 2020). (2) *China three staple crops 1km* product. This product used the GLASS (Global LAnd Surface Satellite) product and a phenology-based method to produce distribution maps of three staple crops in China from 2000 to 2015 at 1-km resolution (Luo et al., 2020). (3) *APRA500* product, which produced rice distribution maps covering the entire Asian monsoon region from 2000 to 2021 at 500-m resolution using MODIS imagery and a phenology-based method (Han et al., 2022). (4) *Heilongjiang rice map* product, which produced rice distribution maps of Heilongjiang Province every five years from 1990 to 2020 at 30-m resolution using Landsat imagery and a phenology-assisted machine learning method (Zhang et al., 2023a).



### 2.3 Method

Figure 3 illustrates the flow of the rice mapping method proposed in this study, which consists of the following four steps:

(1) selection of training samples; (2) extraction and preprocessing of training time series; (3) model training and classification;

and (4) post-processing of the results.

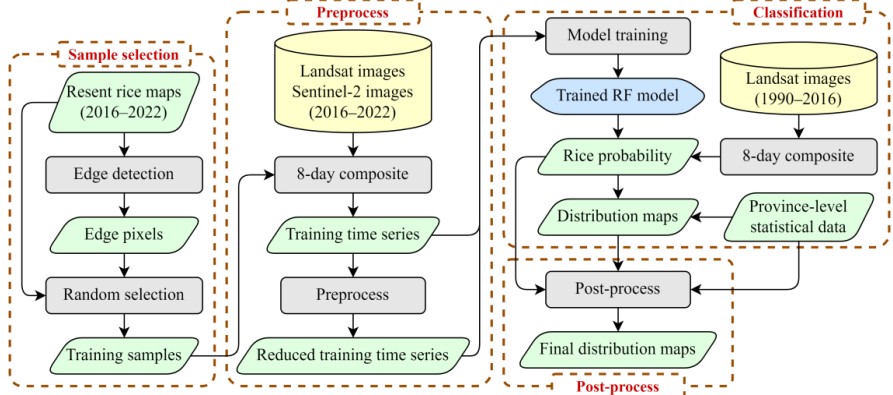

**Figure 3: The conceptual flowchart of the method.**

### 2.3.1 Training sample selection and preprocessing

We extracted training samples from two recent rice distribution map products mentioned in section 2.2.2. For provinces

cultivating only single- or double-season rice, this study randomly extracted 5000 rice pixels and 10,000 non-rice cropland

pixels each year from the distribution map. For provinces where both single- and double-season rice were cultivated, this study

randomly extracted 5000 single-season rice pixels, 5000 double-season rice pixels and 5000 non-rice cropland pixels each year

from the distribution map. We further obtained the SWIR1 time series for these pixels on the GEE platform.

Figure 3a–d shows the extracted time series of rice and non-rice crops in four provincial administrative regions in four

subregions. Notably, in Jilin Province of subregion I, the SWIR1 reflectance of rice and non-rice crops differs significantly

during the transplanting period (DOY 97 to 217) (Fig. 4a). The rice time series first decreased and then increased, showing a

"V" shape, while the time series of other crops remained high. In Shanghai City of subregion II, the "V" shape of the rice time

series is not as noticeable as in Jilin Province, with smaller differences between the time series of rice and non-rice crops.

However, during the entire growing period of rice (DOY 97 to 289), the averaged SWIR1 of rice pixels is always lower than

for non-rice crops (Fig, 3b). In Zhejiang and Guangdong of subregions III and IV, the differences between rice and non-rice

crops are even smaller, but the SWIR1 is still relatively lower than in non-rice crops (Fig. 4c and d).

Figures 3e–h show the percentages of good observations of the extracted training time series. As Landsat 8 and 9 and

Sentinel-2 were used to composite the 8-day training time series, the percentage of good observations was relatively high,

averaging 86.38 %, 64.72 %, 56.20 %, and 47.41 % in these four provinces. Considering that the percentage of good
observations was much higher in subregion I than in the other three subregions, and combined with the significant differences
between rice and non-rice crops during the transplanting period, this study used the time series from DOY 97 to 217 as a model
input in subregion I. Since the percentage of good observations in the other three subregions was lower, and the differences
between rice and non-rice crops were more evenly distributed throughout the rice growing season, this study used time series

of the entire growing season as model input in these three subregions, specifically DOY 97 to 289, DOY 73 to 289, and DOY
33 to 289, respectively.

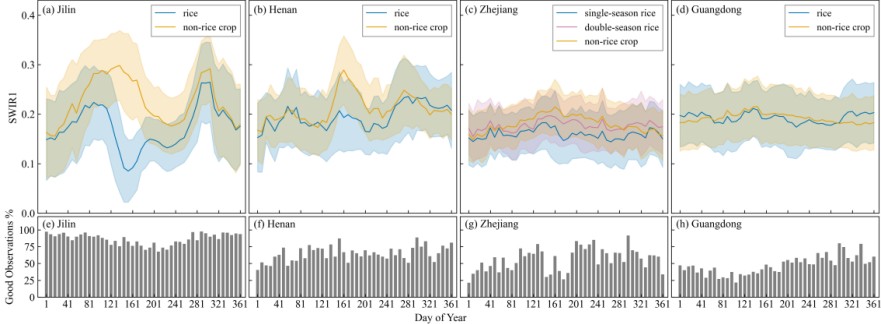

**Figure 4: SWIR1 time series of training pixels and the percentage of good quality observations within the time series for Jilin, Henan, Zhejiang, and Guangdong. Solid lines indicate the average time series, and the shaded error bands represent the standard deviations.**

In addition to these samples, this study also trained the model with some edge pixels to improve its performance. Edge
detection was performed for two recent rice distribution products using the Canny algorithm. Pixels at the edges (i.e., pixels
adjacent to other cover types) were randomly selected for training. The number of selected edge training pixels was one-tenth
of the former pixel sample. Specifically, in provinces where only single- or double-season rice is cultivated, 500 edge rice
pixels and 1000 edge non-rice cropland pixels were extracted annually. In provinces where both single- and double-season rice

were cultivated, 500 edge pixels were extracted annually for each of category (single-season rice, double-season rice, and non-
rice cropland).

During the 1990-2016 study period, most years had observations from only one Landsat satellite, resulting in a lower
percentage of good observations in the time series compared to the time series used for training. Due to differences in the
frequency of good observations between the study years and the training time series, the training time series could not be used

directly in model training. While the traditional solution was to use approaches such as interpolation to fill the gap in the time
series, this study chose to reduce the good quality observation in the training time series. Specifically, this study calculated the
distribution of the percentage of good observations on cropland pixels in each province for the years 1990 to 2016 and
randomly deleted valid observations from the training time series to make the distribution of the percentage of good
observations in the training time series consistent with that for the years 1990 to 2016. Finally, in this study, both the original



training time series and the reduced training time series were used as the model input.

### 2.3.2 Classification method

This study used the RF model from Scikit-learn for image classification (Pedregosa et al., 2011). RF is a versatile decision tree-based classification algorithm. It uses an ensemble of trees, each trained on a subset of the data, to improve accuracy and reduce overfitting. By aggregating predictions through majority voting, RF is effective for a wide range of classification tasks.

In addition to being able to output classification results directly, RF can also provide the probability of each class. For binary classification problems, the class with the highest probability (greater than 0.5) is the model's default classification result. This study used the default RF parameters to train the model for rice classification from 1990 to 2016, using the training data generated in the previous section for each provincial administrative region. This study did not directly use the model output for classification, as the direct output varies greatly from year to year. Instead, we used the probability provided by the RF

model. Specifically, in provinces where only single- or double-season rice is cultivated, we used the rice probabilities provided by the model. Instead of adopting the default rice probability threshold of 0.5, we used the statistical rice planting area of the province for the year to determine a new threshold of rice probability. The total area of pixels with a rice probability greater than the threshold was consistent with the province-level statistical area, and these pixels were identified as rice. In provinces with both single- and double-season rice, we transformed the multi-classification problem into two binary classification

problems. First, we classified rice and non-rice crops, and then we classified single- and double-season rice. Again, the province-level statistical area was used to determine the probability threshold to obtain the classification maps. Note that two rice map products in recent years did not include Tianjin and Hebei due to low rice planting areas. Therefore, this study applied the rice classification model developed for Liaoning Province, which has a similar cultivation context to the two provincial administrative regions. In addition, in some years there were a few pixels that had no high-quality observations during the

study period. Given that the model we trained was applied to all years, the probabilities of the model outputs were to some extent comparable between years. Therefore, we filled these pixels with the probabilities of the neighboring years and then used the threshold to generate the rice maps after the filling was complete.

### 2.3.3 Post-processing of distribution maps

To eliminate fragmented pixels in the classification results, post-processing of the results was performed. Unlike dryland

crops such as maize or soybean, which may rotate, once a plot of land is converted to paddy, it is typically used to cultivate rice over a long time. Therefore, isolated pixels that are classified as rice only a few years are likely to be inaccuracies, as they do not conform to the planting norm. This study set a threshold of five years, overlayed the preliminary rice maps to get the rice planting years on each pixel, eliminated the pixels with planting years less than five years, and regenerated the rice maps

on the remaining pixels using the province-level planting area and the probability of the model output. However, for the first

four years (1990 to 1993) and the last four years (2013 to 2016), some drylands may have been converted to paddy fields or

paddy fields may have been converted to drylands in that year. These new paddy fields could be planted with rice, but could

not meet the 5-year requirement, so in these years we relaxed the requirement and only required that the planting years be

greater than or equal to the number of years between that year and 1990 or 2016.

Given that the interannual rice area within a county-sized region is typically stable, this study applied a post-process to

refine the results. Specifically, each province was divided into several grids of 1000 pixels by 1000 pixels, and the rice area

within each grid was calculated from each year's result. Then, a low-pass filter (five-year sliding average) was applied to the

time series of rice area within each grid. The filtered rice areas eliminated unreasonable fluctuations in rice area from year to

year. We used the filtered rice areas to redetermine the threshold of rice probability of each year within each grid, following

the method mentioned in section 2.3.2, and regenerated the rice maps of each year within each grid using these thresholds.

**2.3.4 Accuracy assessment**

This study used validation samples and agricultural statistical data mentioned in section 2.2.3 to validate the results. We

compared the result with the validation samples and calculated the confusion matrices and three metrics, user's accuracy (UA),

producer's accuracy (PA), and overall accuracy (OA) to evaluate the accuracy of the result. UA indicates the percentage of

correctly classified rice samples among all rice samples, PA indicates the percentage of correctly classified rice samples among

all samples identified as rice, and OA indicates the percentage of correctly classified samples among all samples.

Since the validation sample could not cover too many years, especially before 2002, we further compared the result with

the agricultural statistical data. A linear regression method was used to measure the relationship between the identified area

and the statistical area, and the coefficient of determination ($R^2$) and the relative mean absolute error (RMAE) were calculated

to measure the accuracy. The calculation of the RMAE was as follows:

$$\text{RMAE} = \frac{\sum_{i=1}^{n}|SA_i - IA_i|}{\sum_{i=1}^{n} SA_i} \qquad (1)$$


where $SA_i$ and $IA_i$ are the statistical and identified areas of the $i$th county, respectively. n indicates the number of counties in

the investigated province.

**2.3.5 Sensitivity analysis**

To evaluate the effectiveness of the data preprocessing strategies described in Section 2.3.1, we conducted a sensitivity

analysis in Jiangsu Province, using our current preprocessing strategies as the control group. The following four experimental

groups were designed. Experimental group I used more accurate training sample points. Specifically, we overlayed the six-

year distribution maps from Shen et al. (2023a) and randomly selected training points on pixels identified as rice in all six





years and pixels identified as non-rice in all six years. Pixels identified as a certain type in all six years are less likely to be misidentified, which can be considered as more accurate training samples. Experimental group II did not delete data from the training sample and trained the model directly using the time series on the training points from 2017 to 2022. Experimental groups III–V all filled in missing values in the time series. The filling was done by linear interpolation and the time series were smoothed with a Savitzky-Golay (SG) filter (Savitzky and Golay, 1964). Experimental group III performed the filling directly on the training samples and the time series used for prediction. Experimental group IV, on the other hand, first deleted observations from the training time series using the same method as the control group, and then filled in the missing values in both the training time series and the time series used for prediction. Experimental group V randomly selected 50 time series from the filled rice time series to synthesize standard rice time series and used the TWDTW method described in Shen et al. (2023a) to generate rice distribution maps. All other steps and post-processing for all experimental groups were kept consistent with the control group.

## 3. Results

### 3.1 Accuracy of rice distribution maps

The validation samples were used to verify the accuracy of the distribution maps. The distribution maps achieved high accuracy in almost all provincial administrative regions (Table 2 and 3). Specifically, the user's accuracy, producer's accuracy, and overall accuracy for rice averaged 88.40 %, 89.10 %, and 90.26 %, respectively, across 18 provincial administrative regions where only single- or double-season rice were cultivated. Liaoning had the highest user's accuracy at 98.55 %, while Shanghai had the lowest at 69.69 %. Ningxia had the highest producer's accuracy at 99.40 %, while Hainan had the lowest at 70.00 %. The highest overall accuracy was in Liaoning at 96.63 %, while the lowest was in Sichuan at 78.00 %. For seven provincial administrative regions where both single- and double-season rice were cultivated, the user's accuracy for single- and double-season rice averaged 82.92 % and 81.67 %, respectively, the producer's accuracy for single- and double-season rice averaged 86.45 % and 79.50 %, respectively, and the overall accuracy averaged 88.54 %. The highest overall accuracy was in Hubei at 93.41 %, while the lowest was in Anhui at 82.17 %.

**Table 2: Confusion matrices of the rice distribution maps in 18 provincial administrative regions where only single- or double-season rice were cultivated.**

| Province | Class | Rice[a] | Other | UA (%) | PA (%) | OA (%) |
|---|---|---|---|---|---|---|
| Heilongjiang | Rice[b] | 66083 | 5602 | 98.15 | 92.19 | 93.71 |
| | Other | 1246 | 35969 | 86.52 | 96.65 | |
| Jilin | Rice | 61877 | 2380 | 89.07 | 96.30 | 90.94 |
| | Other | 7595 | 38231 | 94.14 | 83.43 | |
| Liaoning | Rice | 2448 | 237 | 98.55 | 91.17 | 96.63 |



| | | | | | |
|---|---|---|---|---|---|
| | Other | 36 | 5384 | 95.78 | 99.34 | |
| Hebei | Rice | 1407 | 296 | 90.77 | 82.62 | 89.14 |
| | Other | 143 | 2196 | 88.12 | 93.89 | |
| Inner Mongolia | Rice | 2246 | 64 | 87.77 | 97.23 | 92.61 |
| | Other | 313 | 2479 | 97.48 | 88.79 | |
| Ningxia | Rice | 2480 | 15 | 80.47 | 99.40 | 90.99 |
| | Other | 602 | 3754 | 99.60 | 88.79 | |
| Tianjin | Rice | 4023 | 51 | 82.30 | 98.75 | 92.49 |
| | Other | 865 | 7253 | 99.30 | 89.34 | |
| Jiangsu | Rice | 5775 | 926 | 90.29 | 86.18 | 89.72 |
| | Other | 621 | 7730 | 89.30 | 92.56 | |
| Sichuan | Rice | 1399 | 349 | 73.79 | 80.03 | 78.00 |
| | Other | 497 | 1601 | 82.10 | 76.31 | |
| Yunnan | Rice | 2376 | 78 | 92.52 | 96.82 | 94.33 |
| | Other | 192 | 2112 | 96.44 | 91.67 | |
| Chongqing | Rice | 502 | 65 | 95.80 | 88.54 | 93.31 |
| | Other | 22 | 711 | 91.62 | 97.00 | |
| Guizhou | Rice | 821 | 46 | 91.32 | 94.69 | 93.53 |
| | Other | 78 | 972 | 95.48 | 92.57 | |
| Henan | Rice | 1241 | 212 | 82.40 | 85.41 | 89.26 |
| | Other | 265 | 2725 | 92.78 | 91.14 | |
| Shanghai | Rice | 9325 | 1989 | 69.69 | 82.42 | 84.78 |
| | Other | 4055 | 24341 | 92.45 | 85.72 | |
| Shaanxi | Rice | 2703 | 279 | 94.74 | 90.64 | 91.61 |
| | Other | 150 | 1981 | 87.65 | 92.96 | |
| Shandong | Rice | 3109 | 265 | 91.20 | 92.15 | 94.09 |
| | Other | 300 | 5892 | 95.70 | 95.16 | |
| Guangdong | Rice | 4406 | 1127 | 83.27 | 79.63 | 92.61 |
| | Other | 885 | 20795 | 94.86 | 95.92 | |
| Hainan | Rice | 441 | 189 | 90.00 | 70.00 | 79.73 |
| | Other | 49 | 495 | 72.37 | 90.99 | |

[a] number of visually interpreted samples. [b] number of identified samples.

**Table 3: Confusion matrices of the rice distribution maps in seven provincial administrative regions where both single- and double-season rice were cultivated.**


| Province | Class | SR[a] | DR | Other | UA (%) | PA (%) | OA (%) |
|---|---|---|---|---|---|---|---|
| Hunan | SR[b] | 593 | 16 | 22 | 73.57 | 93.98 | |
| | DR | 80 | 647 | 131 | 83.70 | 75.41 | 85.64 |
| | Other | 133 | 110 | 1693 | 91.71 | 87.45 | |
| Jiangxi | SR | 476 | 57 | 52 | 84.85 | 81.37 | |
| | DR | 48 | 1286 | 113 | 90.31 | 88.87 | 90.20 |
| | Other | 37 | 81 | 1811 | 91.65 | 93.88 | |
| Hubei | SR | 925 | 11 | 132 | 91.49 | 86.61 | 93.41 |





| | | | | | | | |
|---|---|---|---|---|---|---|---|
| | DR | 49 | 93 | 27 | 75.61 | 55.03 | |
| | Other | 37 | 19 | 2877 | 94.76 | 98.09 | |
| | SR | 685 | 104 | 87 | 76.37 | 78.20 | |
| Anhui | DR | 1 | 424 | 24 | 71.50 | 94.43 | 82.17 |
| | Other | 211 | 65 | 1158 | 91.25 | 80.75 | |
| | SR | 1820 | 44 | 148 | 91.37 | 90.46 | |
| Zhejiang | DR | 25 | 816 | 121 | 83.52 | 84.82 | 91.26 |
| | Other | 147 | 117 | 3649 | 93.13 | 93.25 | |
| | SR | 423 | 0 | 31 | 72.68 | 93.17 | |
| Guangxi | DR | 62 | 1524 | 115 | 96.70 | 89.59 | 92.39 |
| | Other | 97 | 52 | 2389 | 94.24 | 94.13 | |
| | SR | 712 | 19 | 144 | 90.13 | 81.37 | |
| Fujian | DR | 23 | 147 | 45 | 70.33 | 68.37 | 84.71 |
| | Other | 55 | 43 | 964 | 83.61 | 90.77 | |

[a] number of visually interpreted samples. [b] number of identified samples. SR and DR mean single- and double-season rice, respectively.

Compared with county-level statistical data, the distribution maps also achieved high performances in both single- and double-season rice. Specifically, the identified area of single-season rice in this study showed a strong linear correlation with the statistical area, with scatters all close to the 1:1 line in all years included in the comparison (Fig. 5). The slopes of the regression lines between identified and statistical areas ranged from 0.87 to 1.09, with an average slope of 1.01, and the $R^2$ values ranged from 0.67 to 0.92, with an average of 0.84. The distribution maps also accurately represent the spatial variation of double-season rice. There are strong linear correlations between the identified double-season rice area and the county-level statistical area for all years (Fig. 6). The slopes ranged from 0.61 to 1.06, with an average of 0.85, and the $R^2$ values ranged from 0.64 to 0.90, with an average of 0.78. In addition, our products also reflect the temporal variation in rice planting area at the provincial level, especially in provinces with large temporal variations (Fig. S1).



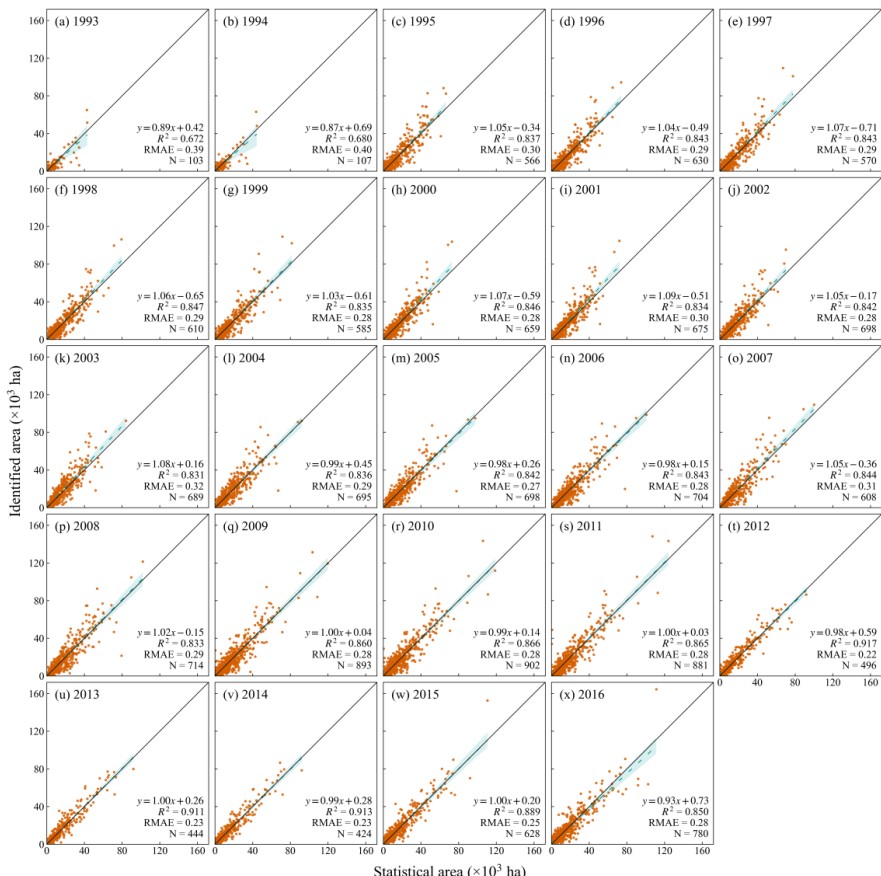

**Figure 5: Comparison between identified single-season rice area and county-level statistics for each year. Solid lines are 1:1 lines and dashed lines are regression lines. The confidence intervals are shaded in blue. N indicates the number of counties included in the comparison. Years with N less than 100 are omitted here.**


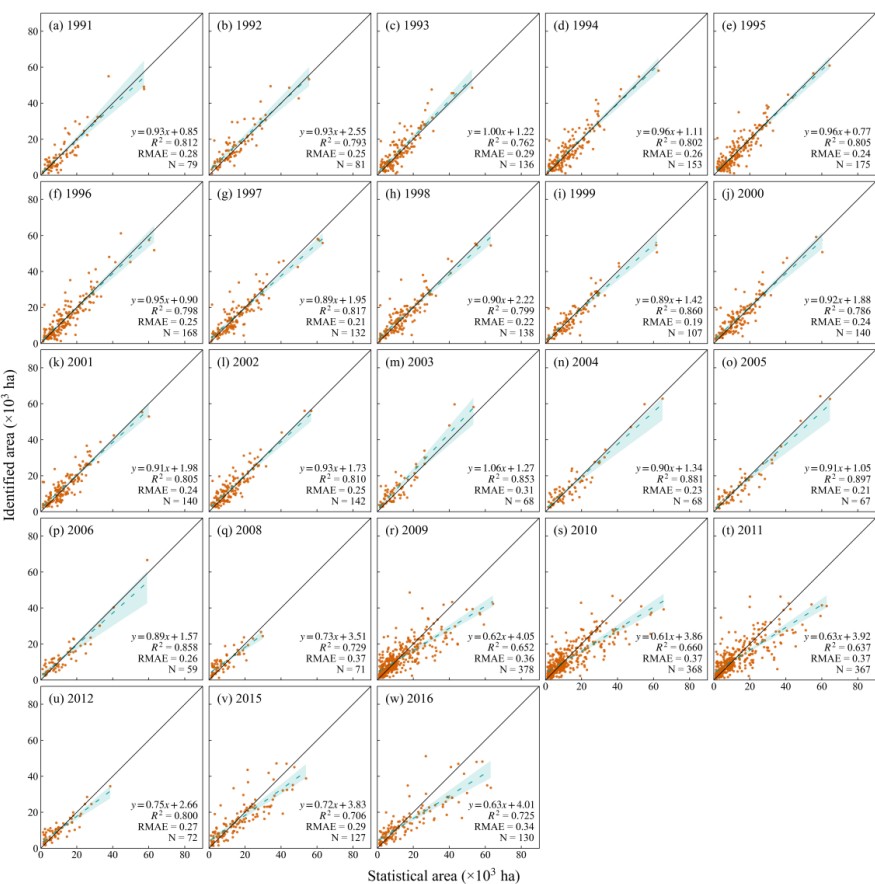

**Figure 6: Comparison between identified double-season rice area and county-level statistics for each year. Solid lines are 1:1 lines and dashed lines are regression lines. The confidence intervals are shaded in blue. N indicates the number of counties included in the comparison. Years with N less than 50 are omitted here.**

The distribution maps achieved high accuracy in all provincial administrative regions. For single-season rice, the average slopes of all years in each province ranged from 0.48 to 1.34, the averaged $R^2$ values ranged from 0.22 to 0.93, and the averaged RMAE ranged from 0.16 to 0.66 (Fig. 7). For double-season rice, the accuracies were slightly lower than those for single-season rice. The average slopes in each province ranged from 0.48 to 0.92, the $R^2$ values ranged from 0.33 to 0.90, and RMAE ranged from 0.19 to 0.48 (Fig. 8).



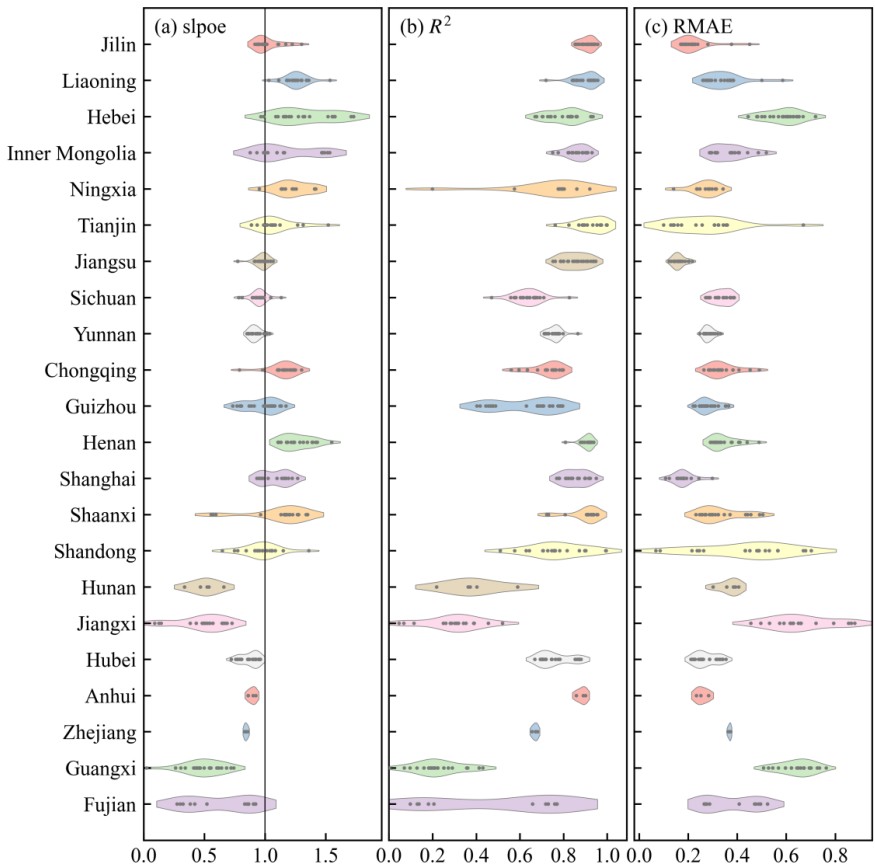


**Figure 7: Comparison between identified single-season rice planting area and county-level statistics by provincial administrative region each year.**

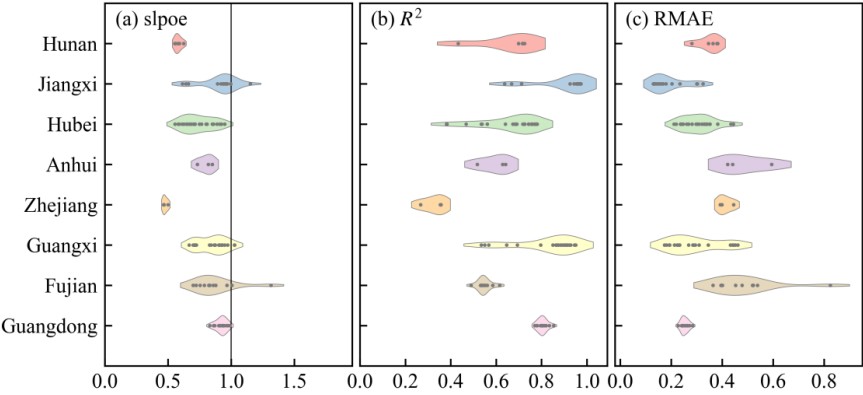

**Figure 8: Comparison between identified double-season rice planting area and county-level statistics by provincial administrative**

**region each year.**





### 3.2 Planting frequency

In this study, the rice maps produced for 25 provincial administrative regions in mainland China from 1990 to 2016 accurately reflect the distribution of rice cultivation in China during the 27-year period (Fig. 9). Rice cultivation in the Northeast, Yangtze-Huaihe, and Southwest was dominated by single-season rice, and the planting frequency is lower than that

in the Southeast provinces, where double-season rice is cultivated. The lowest average planting frequency was 11.21 in Chongqing, and the highest average planting frequency was 30.89 in Jiangxi (Fig. 9). For single-season rice, the highest average planting frequency of single-season rice was 19.31 years, in Liaoning, and the lowest was in Guangxi, only 5.21 years (Fig. 10). For double-season rice, Jiangxi was the province with the highest planting frequency, at 14.29 years, while Anhui had the lowest planting frequency, at 7.24 years (Fig. 11).

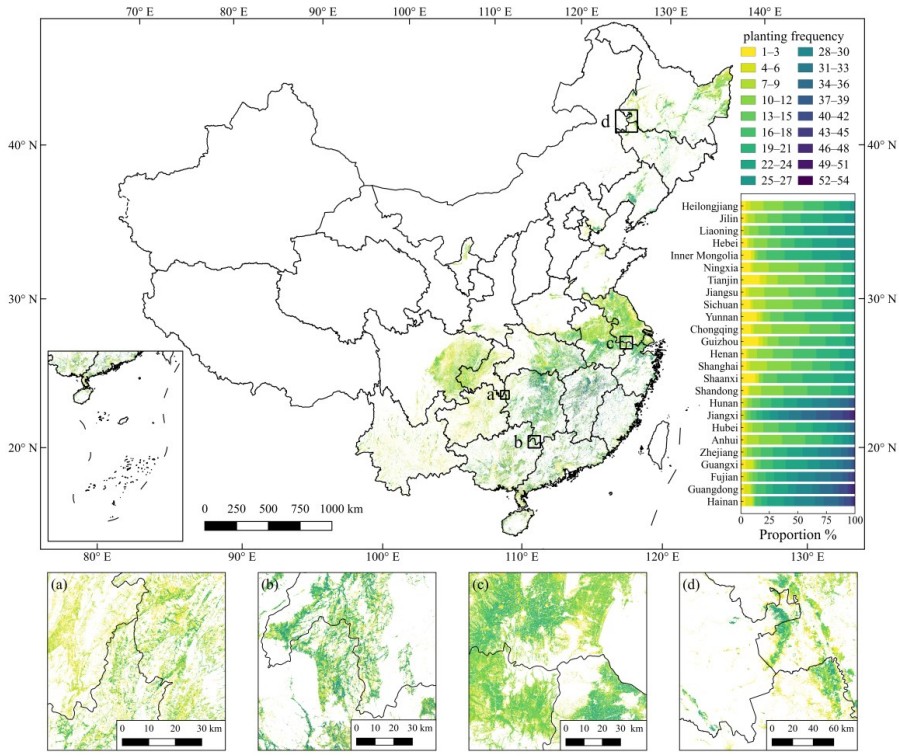


**Figure 9: Planting frequency of rice from 1990 to 2016. Panels a–d on the bottom are the zoomed-in maps.**



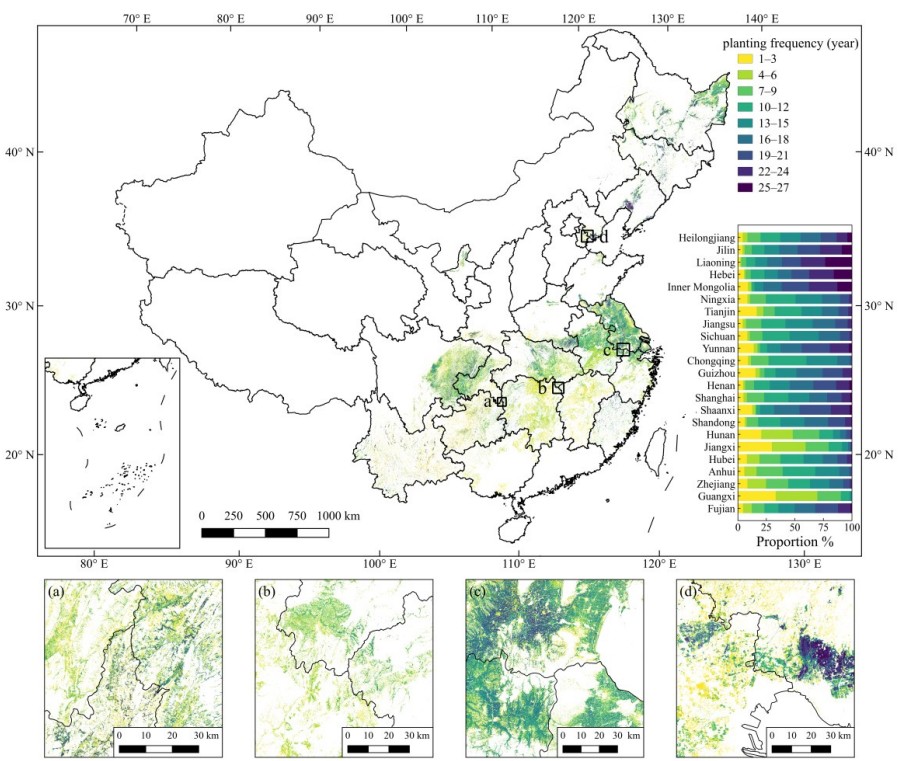

**Figure 10: Planting frequency of single-season rice from 1990 to 2016. Panels a–d on the bottom are the zoomed-in maps.**

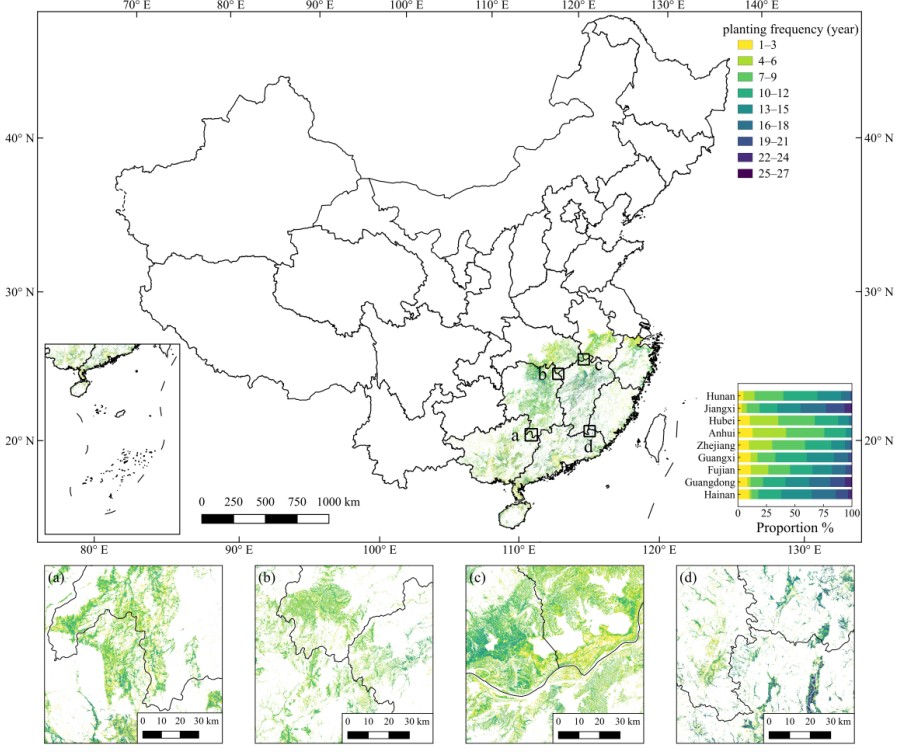

**Figure 11: Planting frequency of double-season rice from 1990 to 2016. Panels a–d on the bottom are the zoomed-in maps.**

### 3.3 Comparison with existing products

To demonstrate the ability of our products to depict the details of rice fields, very high-resolution images obtained from Google Earth were used to compare the actual distribution of rice with the distribution map in four small areas in Heilongjiang, Jilin, Shanghai, and Guangdong. The images were taken in 2010, 2007, 2004, and 2010, respectively. This study compared

the four existing products mentioned in section 2.2.4 on these four small areas (Fig. 12). To facilitate the comparison, we visually interpreted the images and labeled the rice fields (Fig. 12a2–d2). The result showed high performance in all four small areas, accurately reflecting rice cultivation patterns (Fig. 12a3–d3). The *NEAsia_Rice* product was able to roughly reflect the distribution of rice cultivation in both small areas but was limited in its ability to portray the details of paddy fields due to its spatial resolution (Fig. 12a4–b4). The *China three staple crops 1km* product differs significantly from the actual rice field

distribution in all four small areas (Fig. 12a5–d5). The *APRA500* product roughly reflects the rice planting distribution in the first three study areas but fails to do so in the fourth area (Fig. 12a6–d6). In contrast, the *Heilongjiang rice map* product provides a detailed portrayal of rice field distribution (Fig. 12a7).

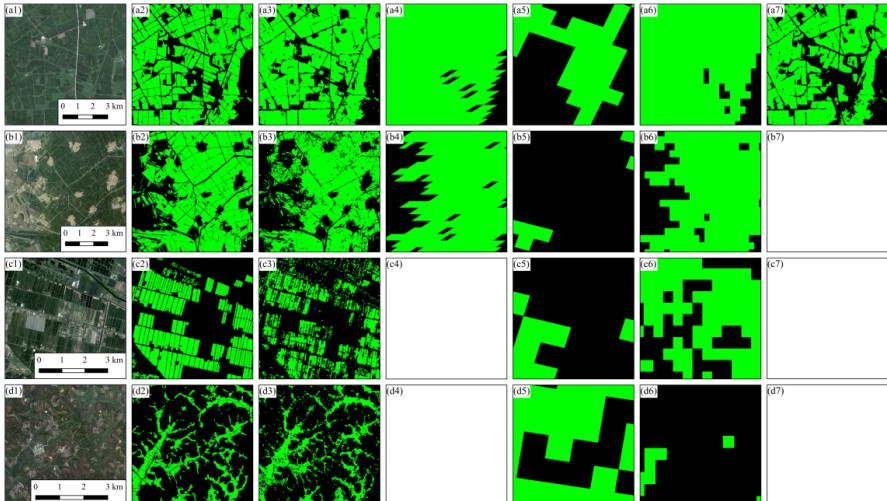

**Figure 12: Comparison of this study with four other studies on four small areas located in Heilongjiang (45°52′11″ N, 132°52′16″ E),**

**Jilin (45°17′52″ N, 124°37′9″ E), Shanghai (31°42′46″ N, 121°28′24″ E), and Guangdong (21°25′30″ N, 110°36′0″ E), respectively. The first column shows very high-resolution imagery obtained from © Google Earth, with image acquisition dates of 24 July 2010, 11 June 2007, 21 July 2004, and 2 September 2010. The second column shows visually interpreted results. The third to seventh columns show the classification maps from this study, the NEAsia_Rice product, the China three staple crop 1km product, the APRA500 product, and the Heilongjiang rice map product, respectively. Blank panels indicate that the product did not have a**

**classification map for that area.**

In addition to the higher spatial resolution, the accuracy of the distribution maps of this study was also superior to that of existing products. We validated the existing products using statistical data on rice field area, which is calculated as the sum of



the planting area of single- and double-season rice, as not all products distinguished between single- and double-season rice.

However, the Heilongjiang rice map product could not be validated due to the unavailability of statistics in Heilongjiang.

Compared with the statistical rice planting area, the distribution maps of this study had a higher $R^2$ value and a lower RMAE

than three existing products in most provinces and years (Fig. 13). Specifically, the $R^2$ values of the maps from this study with

statistical data were higher than the *NEAsia_Rice* product in 70.59 % of the years and provinces, higher than the *China three*

*staple crop 1km* product in 93.44 % of the provinces and years, and higher than the *APRA500* product in 92.41 % of the years

and provinces. Meanwhile, the RMAE of this study is lower than the *NEAsia_Rice* product in all years and provinces, lower

than the *China three staple crop 1km* product in 95.75 % of provinces and years, and lower than the *APRA500* product in

97.93 % of years and provinces.

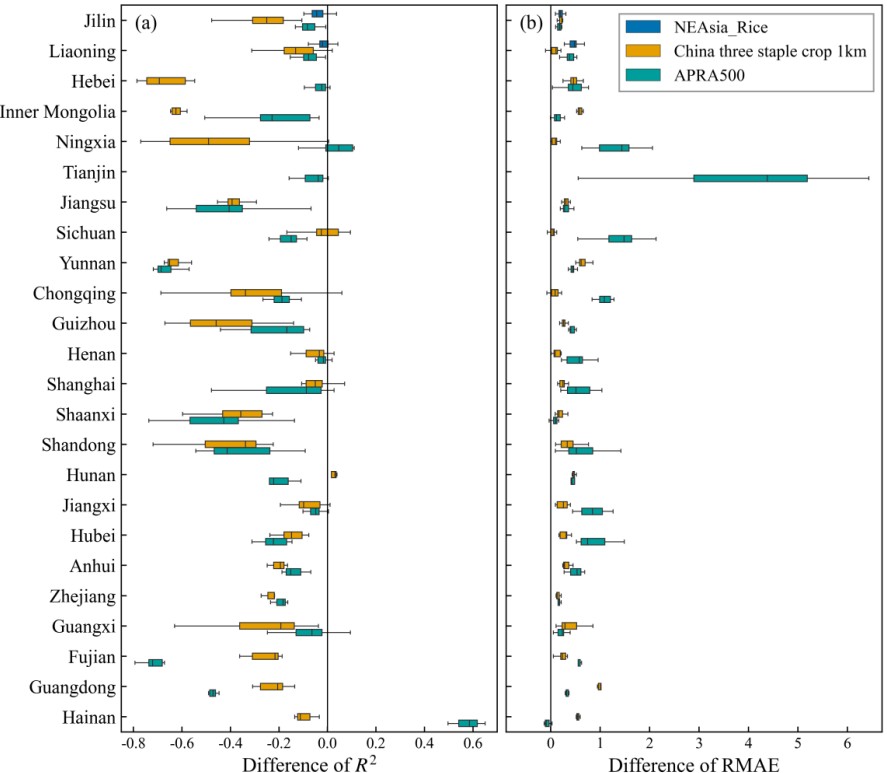

**Figure 13: Differences in $R^2$ and RMAE of the comparison with the statistical data between three exist maps and our distribution maps, respectively.**

Additionally, we compared the accuracy of our map with that of existing products using validation samples. We only

compared our map with the *Heilongjiang rice map* product due to the lower spatial resolution of the other products, which

made them unsuitable for validation with 30 m resolution samples. The comparison in Heilongjiang in 2010 showed that our

map achieved similar accuracy to the *Heilongjiang rice map* product (Table 4). The UA of our map was higher, while PA and





OA were slightly lower than those of the *Heilongjiang rice map* product.

**Table 4: Confusion matrices of the distribution map of our map and *Heilongjiang rice map* product in Heilongjiang 2010.**

| Product | Class | Rice[a] | Other | UA (%) | PA (%) | OA (%) |
|---|---|---|---|---|---|---|
| Our map | Rice[b] | 58064 | 5373 | 98.83 | 91.53 | 93.06 |
| | Other | 687 | 23251 | 81.23 | 97.13 | |
| *Heilongjiang rice map* | Rice | 55221 | 1661 | 93.99 | 97.08 | 94.06 |
| | Other | 3530 | 26963 | 94.20 | 88.42 | |

[a] number of visually interpreted samples. [b] number of identified samples.

## 4. Discussion

### 4.1 Superiority of CCD-Rice dataset

Although rice is one of the most important crops in China, few long-term rice map products are available due to various

challenges. The main difficulties in long-term rice mapping stem from two factors: mapping methods and quality of remote

sensing data. Machine learning methods require a large volume of training samples; transferring the model between years is

an additional challenge (Belgiu and Csillik, 2018; Millard and Richardson, 2015). The most accurate approach is to collect

samples for model training every year. However, the precise planting situations in past years are difficult to collect through

field surveys, even if farmers are asked about their past plantings. On the other hand, knowledge-based methods such as

phenology-based approaches may require no or very little training data. However, while these methods are not limited by

transferability between years, they are more affected by the quality of observations (Dong et al., 2016; Shen et al., 2023a).

Southern China is cloudier and rainier, resulting in less reliable observations of optical remote sensing data (Li and Chen,

2020). High spatial resolution satellites, such as Landsat, typically have lower temporal resolutions, which can exacerbate the

effects of missing data (Li et al., 2024). This data limitation hinders the application of both methods, especially for phenology-

based methods that rely on irrigation signals during the transplanting period. Therefore, existing high-resolution long-term rice

distribution maps were limited to less cloudy and rainy areas such as Northeastern China (You et al., 2021; Zhang et al., 2023a).

Medium-resolution optical satellites usually have high temporal resolution and can largely reduce the probability of not having

cloud-free observations during rice transplantation. As a result, many studies use medium-resolution optical satellites to

produce rice distribution products (Han et al., 2022; Luo et al., 2020; Xiao et al., 2005; Zhang et al., 2017). However, their

low spatial resolution precludes these products from accurately depicting the details of rice cultivation. When compared with

statistical data, the accuracies of these products are lower than those in this study (Fig. 12 and 13).

Although rice differs most from other crops during the transplanting period, there are spectral characteristics that

distinguish it from other crops during other stages of rice growth (Fig. 4). These characteristics have been observed in some

previous studies, and some of the studies have also utilized images of the entire growing season (Shen et al., 2023a; Xuan et al., 2023; Zhang et al., 2023a). This study also utilized remote sensing images throughout the entire growing season of rice in southern China, rather than just during the transplanting period, resulting in more usable images for rice classification. Such a strategy allowed this study to achieve high-resolution rice mapping in southern China using only Landsat data.

In conclusion, previous studies have failed to achieve long-term, high-resolution rice mapping in China. Compared with previous products, our product has the advantages of wide coverage (all of China), high resolution (30 m), long-term (27 years),

and differentiation of cropping systems (single- and double-season rice). Furthermore, this product contributes to the China Crops Dataset (CCD), following CCD-Maize and CCD-Wheat (Dong et al., 2020, 2024; Peng et al., 2023; Shen et al., 2022). Together, the three datasets form a long-term, high-resolution distribution dataset of the three major staple crops in China, providing crucial data support for crop research in China.

### 4.2   Sensitivity analysis of the classification method

In this study, unique strategies for training sample selection and preprocessing differed from common practice. Specifically, to overcome the limitation of insufficient training samples required for machine learning methods, this study obtained a large volume of training samples from two recent rice maps mentioned in section 2.2.2. The samples obtained from the recent rice maps are more evenly distributed in the study area than those from the field surveys, and cover both rice and non-rice land covers throughout the region. In this study, valid data were randomly deleted from the training data to simulate

the effect of cloud contamination on the observations and improve the ability of the model to be transferred to previous years. However, it has not yet been demonstrated whether these two strategies are effective. Firstly, there is some uncertainty in the two recent rice maps, and random sampling may result in many mistakenly labeled training samples. This study also selected some training samples at the edges of the rice maps, which may include erroneous data. Additionally, instead of filling the missing values in the time series through interpolation as done in previous studies, valid observations were randomly deleted.

However, it is unclear whether this deletion strategy effectively aids in the model's transferability to other years.

To test the effectiveness of our current preprocessing strategies, we designed several experiments, as elaborated in section 2.3.5, and validated the identification results for each experimental group using county-level statistical data. The average $R^2$ for the control group and for the five experimental groups were 0.85, 0.75, 0.49, 0.74, 0.81, and 0.56, respectively (Fig. 14a). The average RMAE for the control group and the five experimental groups were 0.16, 0.21, 0.40, 0.25, 0.20, and 0.26,

respectively (Fig. 14c). In almost all years, the control group had the highest $R^2$ and the lowest RMAE (Fig. 14b and d). The results of validation using the validation sample were the same. The overall accuracy of the control group was higher than that of any of the experimental groups (Table 5).

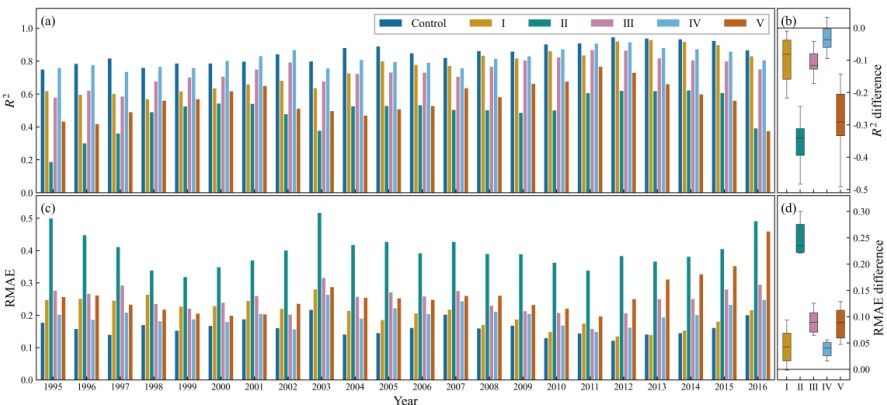

**Figure 14: Comparison between identified single-season rice planting area and county-level statistics of the control group and five experimental groups each year. Panels a and c are the $R^2$ and RMAE of the comparison, respectively. Panels b and d are the differences in $R^2$ and RMAE between the five experimental groups and the control group, respectively.**

**Table 5: Confusion matrices of the distribution maps of the control group and five experimental groups.**

| Groups | Class | Rice[a] | Other | UA (%) | PA (%) | OA (%) |
|---|---|---|---|---|---|---|
| Control group | Rice[b] | 5775 | 926 | 90.29 | 86.18 | 89.72 |
|  | Other | 621 | 7730 | 89.30 | 92.56 |  |
| Experimental group I | Rice | 5778 | 1110 | 90.34 | 83.89 | 88.52 |
|  | Other | 618 | 7546 | 87.18 | 92.43 |  |
| Experimental group II | Rice | 2719 | 1842 | 42.51 | 59.61 | 63.33 |
|  | Other | 3677 | 6814 | 78.72 | 64.95 |  |
| Experimental group III | Rice | 4714 | 1113 | 73.70 | 80.90 | 81.43 |
|  | Other | 1682 | 7543 | 87.14 | 81.77 |  |
| Experimental group IV | Rice | 5087 | 1283 | 79.53 | 79.86 | 82.78 |
|  | Other | 1309 | 7373 | 85.18 | 84.92 |  |
| Experimental group V | Rice | 2238 | 1043 | 34.99 | 68.21 | 65.45 |
|  | Other | 4158 | 7613 | 87.95 | 64.68 |  |

[a] number of visually interpreted samples. [b] number of identified samples.

Some previous studies have adopted strategies to improve the accuracy of training samples during the selection process (Wen et al., 2022; Zhang and Roy, 2017). Some of these strategies are: selecting only pixels that remain constant across years, selecting only pixels whose neighboring pixels are all the same type, or selecting only pixels at the center of patches. These strategies improve the accuracy of the samples to a great extent and avoid including erroneous samples. However, this sampling method may reduce sample diversity to some extent. Pixels that have undergone land cover changes or are situated at the edges are excluded from model training, which weakens the ability of the model for such pixels. There are also some studies that suggest that more diverse samples help to improve the accuracy of the model when selecting training samples (Fu et al., 2023). The comparison with Experimental Group I indicates that more diverse training samples improve the performance of the

classification model (Fig. 14 and Table 5). This improvement may be because pixels located at the image edges are more likely to have features in the feature space that are close to the classification decision boundary.

Time series analysis generally requires complete series. Previous studies typically perform gap filling and filtering to preprocess time series of remote sensing images. This study diverged from previous studies by adding missing values into the time series. Comparisons between the control and experimental group II, as well as between experimental groups III and IV, demonstrate that adding missing values to the time series indeed improves model performance (Fig. 14 and Table 5). This improvement is attributed to the composite training time series of recent years using Landsat and Sentinel-2 data, which have significantly fewer missing values compared to previous years. The model trained with such training data could not make correct predictions for past time series with more missing values. The results of the control group comparing experimental groups III and IV show that using the time series after filling in the missing values resulted in lower accuracy (Fig. 14 and Table 5). Interpolation methods estimate missing observations based on the available data in a time series. However, when the time series contains rapid and transient signals, such as the flooding signal during rice transplanting—which may last only a few weeks—the reliability of these estimates is significantly compromised. According to the Nyquist-Shannon sampling theorem, aliasing occurs when the sampling frequency is insufficient, making it impossible to reconstruct the original signal from the sampled data. In such cases, interpolation not only fails to provide reliable information but also introduces errors into the classification model, ultimately degrading its performance. A recent study also found that using interpolation to fill in missing values in time series does not increase the accuracy of classification models (Che et al., 2024). The comparison with the experimental group V demonstrates that the phenology-based rice classification method is not applicable in the case of poor optical observations (Fig. 14 and Table 5).

### 4.3 Uncertainties

The model results were post-processed. Pixels with no good optical observation during the study period were filled with values from neighboring years. For most years and most provinces, the percentage of filled pixels was less than 1 % (Fig. 15). In several years in Guizhou, Chongqing, and Sichuan, the quality of the optical observations was poor, and the percentage of filled pixels was high, exceeding 5 %, which would increase the error of the product to some extent (Fig. 15). Paddy fields do not have the same flexibility to grow a wide range of crops as drylands, so filling with results from neighboring years is considered a desirable solution.

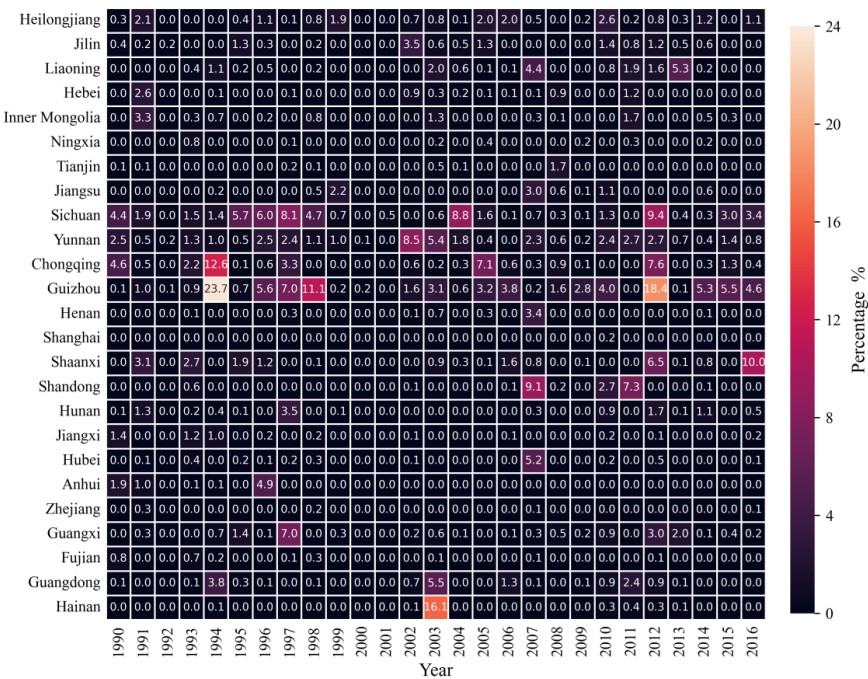

**Figure 15: Percentage of filled pixels to cropland pixels in each year in each provincial administrative region.**

Rice mapping research has historically been constrained by the quality of optical remote sensing data. In this study, a new rice mapping method was developed to improve the temporal transferability of the classification model through suitable preprocessing and to enhance the robustness of the classification model against missing values in the time series. However, the method used in this study is still relatively simple and does not truly enable the model to understand the missing values in the time series. Several studies have pointed out that some deep-learning methods yield better results when handling time

series data with missing values (Che et al., 2018). In addition, the method does not completely solve the influence of low-quality optical remote sensing data. The preliminary products still had regional heterogeneity due to variations in data quality across different areas, which also led to interannual anomalous fluctuations in the rice area in the product. Consequently, this study has undertaken further post-processing to mitigate these fluctuations. Additionally, a small proportion of pixels with zero good observations that need to be filled with neighboring years, which introduces uncertainty into the results. Many

recently developed data fusion methods can combine the advantages of multi-source remote sensing data to provide more reliable time series with valid information for crop classification (Li et al., 2024; Meng et al., 2024). We hope that these advances will further address the limitations of optical remote sensing data quality and produce more accurate rice classification products.

### 5. Data availability

The distribution maps of rice in China from 1990 to 2016 (CCD-Rice) are publicly available on https://doi.org/10.57760/sciencedb.15865 (Shen et al., 2024a). The file format of the product is GeoTIFF with the spatial reference of WGS84 (EPSG:4326). Alternatively, the distribution maps can be viewed through a Google Earth Engine App using the following link: https://ee-shenrq.projects.earthengine.app/view/ccd-rice. The validation samples are available on https://doi.org/10.6084/m9.figshare.25515019.v1 (Shen et al., 2024b). The file format of the validation samples is GeoParquet,

and the geometries are Polygons.

### 6. Code availability

     The codes used to produce the CCD-Rice product is publicly available on https://github.com/shenrq/CCD-Rice.

### 7. Conclusions

     In this study, a new optical satellite-based rice mapping method was developed by combining a machine learning model

with appropriate data preprocessing strategies to address the challenges of cloud contamination and missing data in optical remote sensing observations. Using this method, this study produced the first long-term (1990–2016), high-resolution (30 m) paddy rice distribution dataset in China. The distribution maps captured the spatiotemporal changes of single- and double-season rice cultivation across 25 provincial administrative regions in mainland China. Validation using 391,659 validation samples and 20,759 agricultural statistical records showed high accuracy, with an average overall accuracy of 90.26 % and

strong correlations between mapped and statistical areas, with an average $R^2$ of 0.84 and 0.80 for single- and double-season rice, respectively. This study also demonstrated the validity of the methodology by comparing different preprocessing strategies, including training sample selection strategies, and missing value filling strategies in the time series. Overall, the distribution maps produced in this study demonstrate good accuracy and provide a comprehensive and reliable dataset for monitoring long-term changes in rice cultivation in China, and provide strong data support for food security, sustainable

agriculture, and other related studies.

### Author contributions

     RS and WY conceptualized the study. RS and QP performed the investigation. RS and XL developed the method. RS implemented the computer code, performed the formal analysis, validation, and visualized the results, and wrote the manuscript. WY, XC, and QP edited and revised the manuscript.



**Competing interests**

The authors declare that they have no conflict of interest.

**Financial support**

This study was supported by the National Natural Science Foundation of China (42141020).

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
