# Peer review of "CCD-Rice: A long-term paddy rice distribution dataset in China at 30 m resolution"

_Earth System Science Data, 2024_

## Author Comment (AC1)

MS No.: essd-2024-584

MS Type: Data description paper

Title: CCD-Rice: A long-term paddy rice distribution dataset in China at 30 m resolution

Dear referees and editor,

We are very grateful for the constructive comments and suggested amendments on our manuscript "CCD-Rice: A long-term paddy rice distribution dataset in China at 30 m resolution" (MS No.: essd-2024-584). We have carefully studied the comments, and revised our manuscript accordingly. Consequently, our manuscript has been considerably improved.

Our detailed responses to the comments are in the supplement. Please note that the comments and the relevant contexts are in **bold**, followed by our responses in regular text. The revised and newly added sentences have been highlighted in red.

Sincerely,

Ruoque Shen, Wenping Yuan, on behalf of all co-authors
Email: yuanwp@pku.edu.cn

**Reply to Referee #1**

**I am very appreciated to give me the chance to review such an interesting article, the longer period, higher resolution, and covering the whole China. Such a valuable dataset will hugely improve the ability to estimate greenhouse gas fluxes, climate change impacts and crop production. The authors developed a machine learning model after taking appropriate data preprocessing strategies (so-called addressing the challenges of cloud contamination and missing data in optical remote sensing observations) to produce CCD-Rice (China Crop Dataset-Rice), from 1990 to 2016, with the overall accuracy of 89.89 % at a provincial administrative region, and 85% and 78% for single- and double-season rice at a county administrative scale. Their study is fallen closely within ESSD, and will potentially attract wider readerships. I am highly concerning on the issue of your samples for training and evaluating. However, I am several concerns listed bellows:**

Thanks for your positive comments. We have revised the manuscript based on the comments.

**1. Training samples: how many training samples were collected? Where are their locations? Is there any time-features (day, month, year etc.) labelled for these training samples. Are they isolated from your validation samples.**

Thank you for your comment. Our training samples were randomly selected from two recent rice distribution datasets produced by Shen et al. (2023) and Pan et al. (2021). Due to the random sampling method, the training samples were randomly distributed across the rice planting areas in each province. The training samples consist of three categories: single-season rice, double-season rice, and non-rice, without any additional labels. The issue of training samples we describe in detail in Section 2.3.1.

"We extracted training samples from two recent rice distribution map products mentioned in section 2.2.2. For provinces cultivating only single- or double-season rice, this study randomly extracted 5000 rice pixels and 10,000 non-rice cropland pixels each year from the distribution map. For provinces where both single- and double-season rice were cultivated, this study randomly extracted 5000 single-season rice pixels, 5000 double-season rice pixels and 5000 non-rice cropland pixels each year from the

distribution map." (Lines 161–165)

The two recent datasets provided distribution maps for single- and double-season rice from 2017 to 2022 and 2016 to 2020, respectively. The validation samples were obtained through visual interpretation of Google Earth imagery from 2002 to 2016. Among them, those from Heilongjiang, Chongqing, Henan, and Guangxi included data from 2016. Since Heilongjiang, Henan, and Chongqing only cultivate single-season rice, there is no overlap with the single-season rice training samples extracted from 2017 to 2022. The only potential overlap may occur with the validation samples from Guangxi in 2016. To ensure complete isolation between the training and validation samples, we removed the 2016 validation samples from Guangxi. The revised number of validation samples and their corresponding accuracies are as follows.

Table 1: Years and number of validation samples in each provincial administrative region

| Province | Years of samples | Number of samples | | |
| --- | --- | --- | --- | --- |
| | | SR | DR | Other |
| Heilongjiang | 2010, 2011, 2016 | 67329 | 0 | 41571 |
| Jilin | 2007, 2015 | 69472 | 0 | 40611 |
| Liaoning | 2011, 2015 | 2484 | 0 | 5621 |
| Hebei | 2008, 2015 | 1550 | 0 | 2492 |
| Inner Mongolia | 2013, 2015 | 2559 | 0 | 2543 |
| Ningxia | 2010, 2015 | 3082 | 0 | 3769 |
| Tianjin | 2002, 2006, 2011, 2014 | 4888 | 0 | 7304 |
| Jiangsu | 2004, 2011, 2013, 2014 | 6396 | 0 | 8656 |
| Sichuan | 2005, 2015 | 1896 | 0 | 1950 |
| Yunnan | 2005, 2013 | 2568 | 0 | 2190 |
| Chongqing | 2005, 2016 | 524 | 0 | 776 |
| Guizhou | 2012, 2015 | 899 | 0 | 1018 |
| Henan | 2010, 2016 | 1506 | 0 | 2937 |
| Shanghai | 2004, 2014 | 13380 | 0 | 26330 |
| Shaanxi | 2005, 2014, 2015 | 2853 | 0 | 2260 |
| Shandong | 2010, 2013 | 3409 | 0 | 6157 |
| Hunan | 2013, 2015 | 806 | 773 | 1846 |
| Jiangxi | 2006, 2012 | 561 | 1424 | 1976 |
| Hubei | 2010, 2015 | 1011 | 123 | 3036 |
| Anhui | 2003, 2013 | 897 | 593 | 1269 |
| Zhejiang | 2003, 2013 | 1992 | 977 | 3918 |
| Guangxi | 2011, 2013 | 573 | 310 | 1149 |
| Fujian | 2013, 2015 | 790 | 209 | 1153 |
| Guangdong | 2010, 2012 | 0 | 5291 | 21922 |

| | Hainan | 2009, 2014 | | 0 | 490 | 684 |
|---|---|---|---|---|---|---|

SR and DR mean single- and double-season rice, respectively.

**Table 3: Confusion matrices of the rice distribution maps in seven provincial administrative regions where both single- and double-season rice were cultivated.**

| Province | Class | SR[a] | DR | Other | UA (%) | PA (%) | OA (%) |
|---|---|---|---|---|---|---|---|
| Hunan | SR[b] | 593 | 16 | 22 | 73.57 | 93.98 | |
| | DR | 80 | 647 | 131 | 83.70 | 75.41 | 85.64 |
| | Other | 133 | 110 | 1693 | 91.71 | 87.45 | |
| Jiangxi | SR | 476 | 57 | 52 | 84.85 | 81.37 | |
| | DR | 48 | 1286 | 113 | 90.31 | 88.87 | 90.20 |
| | Other | 37 | 81 | 1811 | 91.65 | 93.88 | |
| Hubei | SR | 925 | 11 | 132 | 91.49 | 86.61 | |
| | DR | 49 | 93 | 27 | 75.61 | 55.03 | 93.41 |
| | Other | 37 | 19 | 2877 | 94.76 | 98.09 | |
| Anhui | SR | 685 | 104 | 87 | 76.37 | 78.20 | |
| | DR | 1 | 424 | 24 | 71.50 | 94.43 | 82.17 |
| | Other | 211 | 65 | 1158 | 91.25 | 80.75 | |
| Zhejiang | SR | 1820 | 44 | 148 | 91.37 | 90.46 | |
| | DR | 25 | 816 | 121 | 83.52 | 84.82 | 91.26 |
| | Other | 147 | 117 | 3649 | 93.13 | 93.25 | |
| Guangxi | SR | 423 | 0 | 31 | 73.82 | 93.17 | |
| | DR | 62 | 263 | 71 | 84.84 | 55.41 | 85.29 |
| | Other | 88 | 47 | 1047 | 91.12 | 88.58 | |
| Fujian | SR | 712 | 19 | 144 | 90.13 | 81.37 | |
| | DR | 23 | 147 | 45 | 70.33 | 68.37 | 84.71 |
| | Other | 55 | 43 | 964 | 83.61 | 90.77 | |

[a] number of visually interpreted samples. [b] number of identified samples. SR and DR mean single- and double-season rice, respectively.

**2. Accuracy: remote sensing products are generally validated by two ways: sampling points by a confusion matrix, and identified areas by statistics books (both province and county, it is more popular at a smaller administrative scale). Please clearly showcase your results related with two validation methods, and clearly involve them into key-contents sections, such as ABSTRACT.**

Thanks for your suggestion. We have mentioned the accuracy of the two validation methods in our abstract.

"Based on 394,753 validation samples, the overall accuracy of the distribution

maps in each provincial administrative region averaged 89.61 %. Compared with 20,544 county-level statistical data, the coefficients of determination ($R^2$) of single- and double-season rice in each year averaged 0.85 and 0.78, respectively." (Lines 14–16)

**3. Comparison: As conducting your comparison with other open products, it will be more scientific and objective to downscale your resolution into coarser one to align your product with other coarser product; or upscale your resolution into a higher resolution for comparing yours with other refined product.**

Thank you for your comment. In comparison to four existing products, we conducted the following three analyses.

1) We selected four small areas to visualize the mapping results of all products. In this case, we believe that preserving the original spatial resolution of each product more effectively highlights the differences among them.

2) The second comparison used the county-level statistical data. For all products, the planting areas were calculated at the county level, which means the comparison is not related to their original spatial resolution of the products.

3) The third comparison was conducted using the validation sample. We only compared it with the *Heilongjiang rice map* product because our validation samples are at a 30 m resolution, which makes them unsuitable for evaluating other products with medium resolutions.

To summarize, these three comparisons either do not involve the spatial resolution of the product (comparison 2 and 3) or need to demonstrate the advantages of our product in spatial resolution (comparison 1). Therefore, we did not resample the products to the same resolution in comparisons.

**4. It will be better to emphasize your improvements comparing with the recent publications (Shen et al. 2023a, 2023b; Pan et al. 2021a, 2021b). \*\*If no any significant improvement has been made, I think it will be more reasonable to update the related datasets, rather than blindly pursuing the number of publications.\*\***

We appreciate the reviewer's suggestions and fully understand the concerns raised regarding the novelty of our study. However, we would like to emphasize that our

research introduces significant innovations compared to previous studies by Pan et al. (2021) and Shen et al. (2023) in terms of methodology, data sources, and research scope. We would like to highlight the following three key aspects of novelty:

Methodology: This study introduces a novel optical satellite-based approach for rice mapping that utilizes machine learning models combined with carefully designed data preprocessing strategies. In comparison to the phenology-based method (TWDTW) employed in the two previous studies, our approach significantly reduces the impact of cloud contamination and data gaps on mapping accuracy in optical remote sensing observations.

Data Source: We used long-term time series data from the Landsat series, which differs from the Sentinel-1 and Sentinel-2 used in the two previous studies. The extensive observational history of Landsat data enables us to produce long-term (from 1990 to 2016) rice distribution maps in China. This temporal coverage significantly extends that of previous studies.

Rice Cropping Systems: This study maps the distribution of both single- and double-season rice. In contrast, Pan et al. (2021) focused exclusively on double-season rice after 2016, while Shen et al. (2023) were limited to single-season rice after 2017.

Based on these innovative aspects, we believe that our study makes an independent and valuable contribution. In response to the reviewer's suggestion, we have revised the Introduction to include a more detailed sentence discussing the limitations of previous methodologies, which helps to highlight the improvements and innovations introduced in our current study.

"Some studies, such as those by Pan et al., (2021b) and Shen et al., (2023a), have produced nationwide distribution maps of double- and single-season rice in China, respectively. However, due to limitations in the quality of the remote sensing data, both studies covered only recent years (2016–2020 and 2017–2022, respectively). Furthermore, the mapping methods used in these studies were insufficient to achieve high precision rice mapping in years with poor optical remote sensing data quality. To address this gap, this study focuses on mapping rice distribution before 2017 and tackling the challenge of poor-quality remote sensing data. Specifically, this study

intends to (1) develop a new optical satellite-based rice mapping method; (2) produce high-resolution distribution maps of single- and double-season rice in China from 1990 to 2016; (3) evaluate the accuracy of the results and analyze changes in rice cultivation patterns." (Lines 61–68)

**5. Other: As you said in ABSTRACT "This study developed a new optical satellite-based rice mapping method using a machine learning model and appropriate data preprocessing strategies to address the challenges of cloud contamination and missing data in optical remote sensing observations. It will be better to remove "address the challenges of cloud contamination and missing data in optical remote sensing observations" because you have \*\*NOT\*\* input any substantive efforts to improve the image quality or solve the issue of cloud contamination and missing data. Please remove such statement throughout your paper to avoid readers' discontent and protect yourselves.**

Thank you for your suggestion. We have revised the sentence.

"This study developed a new optical satellite-based rice mapping method using a machine learning model and appropriate data preprocessing strategies to mitigate the impact of cloud contamination and missing data in optical remote sensing observations on rice mapping." (Lines 11–13)

**6. Carefully check the manuscript to eliminate the spelling mistakes,e.g. resent in Fig.3**

Thank you for pointing out. We have checked the spelling again and revised Fig. 3.

[Figure]

**Figure 3: The conceptual flowchart of the method.**

**Reply to Referee #2**

**I am appreciated to have a chance to review this submission. Paddy rice is one of the world's most important staple food crops, feeding half of the population. Also, rice fields are one of the main sources of greenhouse gas (GHG) emissions, contributing 12%–26% of global anthropogenic methane emissions. As the world's largest rice producer, mapping the long-term dynamics of paddy rice is critical to food supply security and global climate change mitigation.**

**This study developed a new rice-mapping method with machine learning models to train the remote sensing images throughout the entire growing season of rice, and produce the whole cropping system (single- and double seasons). This product contributes to the China Crops Dataset (CCD)-Rice. Trained and validated with 397,414samples, the overall accuracy of the rice distribution maps at the provincial level was 89.89 %. Their study is within the scope of ESSD, and will be attractive to the readers of the community. However, I have major concerns as below:**

Thanks for your positive comments. We have revised the manuscript based on the comments.

**For the issue of recent crop mapping publications (Shen et al. 2023a, Pan et al. 2021a), I suggest the authors to (1) clarify the CH4 emission and/or crop production for single- and double seasons rice, respectively. (2) If forcing and model available, the contribution of dynamics of single- and double seasons rice to the variations of CH4 emission and/or crop production during 1990-2016. I know this might be difficult due to the data availability and calculation cost. Then the authors could add sentences in the discussion to highlight the scientific value of this new rice mapping study.**

Thank you for your suggestion. Changes in the distribution of rice cultivation lead to changes in yield and methane emissions, which is a significant scientific value of this study. Although we did not conduct a study to estimate yields and methane emissions, we can illustrate the impact of these changes on yields and methane emissions by using statistical data and findings from other research. We have added several sentences in

the discussion section to clarify this point.

"The distribution and cropping systems of rice in China have undergone significant changes in recent decades, resulting in substantial impacts on total rice yield and methane emissions from rice paddies. According to statistical data, from 1990 to 2016, the area of single-season rice increased by 43.25 %, while the area of double-season rice decreased by 43.05 %. Concurrently, the yield of single-season rice increased by 64.78 %, whereas the yield of double-season rice decreased by 35.61 %. These changes in area accounted for more than 80 % of the change in rice yield. Furthermore, research indicates that the northeastward shift in rice cultivation in China has contributed to the declining trend in methane emissions from paddies since 2007 (Ouyang et al., 2023). Consequently, long-term rice mapping is of great scientific significance. However, there are few long-term rice map products are available due to various challenges." (Lines 375–382)

**Minor:**

**1. Technical part: I suggest to emphasize the entire growing season of rice in southern China, rather than just during the transplanting period, resulting in more usable images for rice classification. Compared with previous studies, this study mapping both single-season and double season (much in southern China) rice.**

Thank you for your suggestion. We have revised several sentences in the Discussion to show this point.

"This study also utilized remote sensing images throughout the entire growing season of rice in southern China, rather than just during the transplanting period. This approach not only resulted in more usable images for rice classification but also allowed for the mapping of both single-season and double-season rice without the need to account for differences in transplanting periods between the two types. Compared to previous studies, this strategy allowed this study to achieve high-resolution mapping of both single- and double-season rice in southern China using only Landsat data." (Lines 403–407)

**2. I suggest to rephrase the sentence 'previous studies have failed to achieve long-**

**term, high-resolution rice mapping in China.' or delete it (Line398).**

Thank you for your suggestion, we have deleted the sentence.

**References**

Ouyang, Z., Jackson, R. B., McNicol, G., Fluet-Chouinard, E., Runkle, B. R. K., Papale, D., Knox, S. H., Cooley, S., Delwiche, K. B., Feron, S., Irvin, J. A., Malhotra, A., Muddasir, M., Sabbatini, S., Alberto, Ma. C. R., Cescatti, A., Chen, C.-L., Dong, J., Fong, B. N., Guo, H., Hao, L., Iwata, H., Jia, Q., Ju, W., Kang, M., Li, H., Kim, J., Reba, M. L., Nayak, A. K., Roberti, D. R., Ryu, Y., Swain, C. K., Tsuang, B., Xiao, X., Yuan, W., Zhang, G., and Zhang, Y.: Paddy rice methane emissions across Monsoon Asia, Remote Sensing of Environment, 284, 113335, https://doi.org/10.1016/j.rse.2022.113335, 2023.

Pan, B., Zheng, Y., Shen, R., Ye, T., Zhao, W., Dong, J., Ma, H., and Yuan, W.: High Resolution Distribution Dataset of Double-Season Paddy Rice in China, Remote Sensing, 13, 4609, https://doi.org/10.3390/rs13224609, 2021.

Shen, R., Pan, B., Peng, Q., Dong, J., Chen, X., Zhang, X., Ye, T., Huang, J., and Yuan, W.: High-resolution distribution maps of single-season rice in China from 2017 to 2022, Earth Syst. Sci. Data, 15, 3203–3222, https://doi.org/10.5194/essd-15-3203-2023, 2023.